computational physics/complexity

Granger, causality, transfer entropy, information theory, cryptocurrency

**Author for correspondence:**
Z. Keskin
e-mail: zac.keskin.17@ucl.ac.uk

# Information-theoretic measures for nonlinear causality detection: application to social media sentiment and cryptocurrency prices

## Z. Keskin[1,2] and T. Aste[1,3]

[1]Department of Computer Science & Centre for Blockchain Technologies, and [2]Department of Physics and Astronomy, University College London, Gower Street, WC1E 6EA London, UK
[3]UCL Centre for Blockchain Technologies, University College London, London, UK

TA, 0000-0002-4219-0215

Information transfer between time series is calculated using the asymmetric information-theoretic measure known as transfer entropy. Geweke's autoregressive formulation of Granger causality is used to compute linear transfer entropy, and Schreiber's general, non-parametric, information-theoretic formulation is used to quantify nonlinear transfer entropy. We first validate these measures against synthetic data. Then we apply these measures to detect statistical causality between social sentiment changes and cryptocurrency returns. We validate results by performing permutation tests by shuffling the time series, and calculate the Z-score. We also investigate different approaches for partitioning in non-parametric density estimation which can improve the significance. Using these techniques on sentiment and price data over a 48-month period to August 2018, for four major cryptocurrencies, namely bitcoin (BTC), ripple (XRP), litecoin (LTC) and ethereum (ETH), we detect significant information transfer, on hourly timescales, with greater net information transfer from sentiment to price for XRP and LTC, and instead from price to sentiment for BTC and ETH. We report the scale of nonlinear statistical causality to be an order of magnitude larger than the linear case.

## 1. Introduction

Causality is a central concept in natural sciences, commonly understood to describe where a process, evolving in time, has

some observable effect on a second process. However, the nature of this causative effect is challenging to describe and quantify with precision. There is a long history in determining whether some change truly causes another [1,2], especially when the relationship is not deterministic, and is observed only in aggregate. In this paper, we consider a statistical form of causality, which can be observed in co-dependent time series where a response in the dependent series is more likely to follow after some change in the driving series. The direction of information transfer is forced by requiring the cause to precede the effect. This concept was conceived first by Wiener in 1956 [3], and formalized by Granger in 1969 [4] who was subsequently awarded the Nobel memorial prize in economics for his work on the analysis of time series. In simplest terms, the so-called Granger causality describes the extent to which a given series is better able to be predicted by consideration of the information in prior values of another series. If the response in the dependent series scales as a linear multiple of the driving signal, this relationship is described as a linear coupling. If, instead, the response follows some other function of the signal, the relationship is nonlinear.

In modern portfolio theory, investors commonly calculate correlations between asset types to construct portfolios aiming to maximize their return at a given level of risk [5]. In the search for excess returns, quantitative approaches are often exploited to detect predictive signals across time series. This may be considered as a field of price discovery, which represents a large and fundamental area of financial theory. In the ideal case with complete predictive information, from knowing the movement of one price, the movement of a second can be inferred exactly. In reality information is incomplete and held asymmetrically, and the ability to reliably obtain this and contribute to price formation is generally rewarded by the market. In this paper we explore the effectiveness of two promising techniques for detecting information transfer between time series, applying these to identify causal signals between alternative data and cryptocurrency price returns.

The concept of an entirely peer-to-peer digital currency managed via a distributed ledger was described and applied by Nakamoto in 2008 [6], who named the currency 'Bitcoin'. The proposal and subsequent implementation captured the attention of technologists, economists, libertarians and futurists, and spawned numerous adaptations utilizing the blockchain technology [7], which have come to be known as cryptocurrencies. Trading in these cryptocurrencies has become widely available even to less sophisticated retail investors, and volumes have grown significantly as interest in the currencies has widened. The crypto market is characterized by high volatility which seems to reflect changes in the attitudes of investors. The usage of cryptocurrencies in the traditional economy remains limited, and it is reasonable to assume that prices are in part driven by speculative dynamics, separate to any utility as a medium of exchange or to any revenue-generating process. One factor which contributes to these dynamics is market sentiment, the estimation of which using social media data is an active area of research. We discuss existing literature that identifies signals between social sentiment data and equity prices, and so hypothesize that investor sentiment on future cryptocurrency prices may be expected to feed into the short-term price movements via speculation, which may be detected through the usage of information-theoretic techniques. This paper tests this hypothesis.

The relationship between social media sentiment and price changes has been explored in the literature for traditional markets and, more recently, for crypto markets as well, including by an author of this paper. For instance, Bollen *et al.* [8] showed that the mood of Twitter messages can be used as a proxy for market sentiment, and that this can show a linear relationship with price movements in US equities. Zheludev *et al.* [9] also performed sentiment analysis using natural language processing (NLP) on Twitter data, to show sentiment is significantly coupled with price movements for a number of instruments issued by S&P500 firms. Souza & Aste used Twitter messages to model market sentiment, and showed the nonlinear predictive relationship may be greater than the linear one [10].

In the cryptocurrency market specifically, one of the authors of the present paper has recently applied information-theoretic techniques to approaches from network theory to characterize the structure of the market as a complex system [11]. This provided evidence that the market forms a complex, causally interrelated network linking prices and sentiments across multiple currencies.

Within a linear causality framework, hypothesizing that cryptocurrency price returns depend on prior returns and changes in market sentiment, a Granger causality test can detect the impact of past values of a variable $X_t$ on future values of another variable $Y_t$ [4]. This can be calculated using a vector auto-regressive (VAR) model, which describes the extent to which including past values of $X$ reduces the sum of squared residuals in the regression of $X$ against $Y$, hence estimating the predictive effect, at each lag $k$, of the change in social sentiment at $t - k$ on the returns at time $t$.

The VAR approach performs a regression analysis which is limited to linear associations between variables. To investigate nonlinear relations, we can adopt techniques developed in information

theory. Many popular information-theoretic measures for comparing distributions, such as mutual information, are symmetric and so cannot model a directional information transfer from $X$ to $Y$. Therefore, to generalize Granger causality to the nonlinear case, we adopt the measure formalized by Schreiber [12], known as transfer entropy, which is able to capture the size and also the direction of information transfer.

Transfer entropy arises from the formulation of conditional mutual information; it quantifies the reduction in uncertainty about the dependent time series provided by the past values of the driving signal, conditioning on the past values of the dependent variable itself. This presents a natural way to model statistical causality between variables in multivariate distributions. In the general formulation, transfer entropy is a model-free statistic, able to measure the time-directed transfer of information between stochastic variables, and therefore provides an asymmetric method to measure information transfer. As presented in this paper, transfer entropy appears naturally as a generalization of Granger causality. In fact it has been shown that, for multivariate normally distributed statistics, where the relationship is therefore linear, this is indeed the case: Granger causality and transfer entropy are equivalent [13].

Though developed relatively recently, information-theoretic methods have been used with success in research across disciplines, to detect information transfer where interventionist approaches are not possible. For example, in neuroscience, Vicente *et al.* [14] found transfer entropy to be a superior measure in detecting Granger causality in electrophysiological communication than the auto-regressive formulation. In climatology, Liang derived from first principles a linear information flow measure, and used this to show that El Niño tends to stabilize the Indian Ocean Dipole [15]. This analysis also detected a causal effect in the other direction; the Indian Ocean Dipole was shown to amplify El Niño oscillations. The technique was used with further success by Stips *et al.* [16] to confirm that recent $CO_2$ emissions show a one-way causality towards global mean temperature anomalies, but that on palaeoclimate timescales, this direction is reversed and temperatures drive $CO_2$ levels. Finally, in finance, information transfer was measured between equities indices by Kwon & Yang, showing that the information transfer was greatest from the USA, and greatest towards the Asia-Pacific region [17]. In particular, the S&P500 was shown to be the strongest driver of other stock indices. In an earlier and somewhat related work, Marschinski & Kantz [18] defined and used effective transfer entropy to quantify contagion in financial markets. Similarly, Tungsong *et al.* [19] used transfer entropy to develop upon the previous work by Diebold & Yilmaz [20] in quantifying spillover effects between financial markets, generalizing the methodology and estimating the time evolution of interconnectedness between financial systems.

The rest of the paper is organized as follows. In §2 we provide a brief background on Granger–Geweke causality (the linear causality measure) and transfer entropy (the nonlinear causality measure). In §3 we describe details of the methodology adopted to quantify and validate linear and nonlinear Granger causality, and the techniques used to generate synthetic series of linear and nonlinear causal coupling. Section 4 demonstrates that the methodologies correctly detect statistical causality in the linear and nonlinear case when testing against synthetic data. Results for real data, concerning causality between cryptocurrency price and sentiment, are presented in §5. Section 6 reports conclusions and perspectives.

# 2. Background

We calculate statistical causality between time series using two different approaches. The first approach involves the application of the Granger–Geweke causality test, which assumes linearity and employs vector auto-regressive techniques to estimate the extent to which knowing the driving time series can improve predictions of the dependent series. The second technique requires an estimation of the transfer entropy, which uses conditional mutual information to quantify the predictability of the dependent series. When predictability is increased by considering the past values of the driving variable, statistical causality is observed.

## 2.1. Linear Granger causality

In the linear approach, we model a time series as autoregressive by expressing its value $Y_t$ at time $t$ as a sum of the contributions over $m$ distinct lagged series, using the linear equation

$$Y_t = \sum_{k=1}^{m} \beta_k^{(Y)} Y_{t-k} + \epsilon_t, \tag{2.1}$$

where $\beta_k^{(Y)}$ is a general coefficient term and $\epsilon_t$ is the residual. Linear regression estimates the coefficient parameters $\beta_k^{(Y)}$ which minimize the sum of squared residuals.

To detect whether the values of some second time series $X$ anticipate the future values of $Y$, we can compare equation (2.1) with

$$Y_t = \sum_{k=1}^{m} \beta_k'^{(Y)} Y_{t-k} + \sum_{k=1}^{m} \beta_k'^{(X)} X_{t-k} + \epsilon_t'. \qquad (2.2)$$

We describe the distribution $Y$ as being Granger-caused by $X$ if the residual in the second regression is significantly smaller than the residual in the first. When this holds, then there must be some information transfer from $X$ to $Y$. Following Geweke [21], we can represent the information transfer by

$$TE_{X \to Y} = \frac{1}{2} \log \left( \frac{\mathrm{var}(\epsilon_t)}{\mathrm{var}(\epsilon_t')} \right), \qquad (2.3)$$

where we adopt the transfer entropy notation (TE), following the result from Barnett $et\ al.$ [13] showing Granger causality to be equivalent to transfer entropy for multivariate normal distributions.

## 2.2. Nonlinear Granger causality

To detect nonlinear Granger causality, we apply an information-theoretic approach. Equation (2.3) measures the extent to which the additional information in the lagged variable reduces the variance in the model residuals. Transfer entropy extends this concept by considering the uncertainty, instead of the variance. Adopting Shannon's measure of information [22], we can express the uncertainty associated with the continuous random variable $X$ by

$$H(X) = - \int_{-\infty}^{+\infty} p(x) \log p(x) \, \mathrm{d}x, \qquad (2.4)$$

where $p(x)$ is the probability density function. $H(x)$ is termed the differential entropy, and it quantifies the uncertainty on the continuous variable $X$. It is an extension of the Shannon entropy for continuous variables. In this definition there is an arbitrary additive constant that takes into account the scale factor for the variable $x$, which in this paper we set to zero. This arbitrary scale factor is a fundamental problem in the definition of Shannon entropy for continuous variables.

However, in the calculation of mutual information and transfer entropy, these constant factors cancel out, and results are independent of scale. For discrete variables, where the integral becomes a sum, the constant is equal to zero. Using a logarithm of base 2, the entropy value quantifies the minimum number of bits necessary to encode the signal.

Shannon entropy can be directly extended to the multivariate case, defining the joint entropy of two variables $H(X, Y)$ using the joint probability density function $p(x, y)$ in equation (2.4). Shannon entropy can also be conditioned on a second variable to give the conditional entropy

$$H(Y|X) = H(X, Y) - H(X), \qquad (2.5)$$

which is the residual uncertainty in variable $Y$ after taking account of the information in variable $X$.

The information shared between two variables $X$ and $Y$ is the mutual information

$$I(X; Y) = H(Y) - H(Y|X). \qquad (2.6)$$

Similarly, the entropy of $Y$ conditioned on two variables is

$$H(Y|X, Z) = H(X, Y, Z) - H(X, Z), \qquad (2.7)$$

and the conditional mutual information is therefore

$$I(X; Y|Z) = H(Y|Z) - H(Y|X, Z). \qquad (2.8)$$

Now, for each lag $k$, we can describe the information transfer from $X_{t-k}$ to $Y_t$ in terms of the following conditional mutual information:

$$TE_{X \to Y}^{(k)} = I(Y_t; X_{t-k}|Y_{t-k}) = H(Y_t|Y_{t-k}) - H(Y_t|Y_{t-k}, X_{t-k}). \qquad (2.9)$$

This is the reduction of uncertainty in $Y$ when considering the past values of both $Y$ and $X$, compared with considering the past values of $Y$ alone.

Considering equations (2.5) and (2.7), we can therefore represent the transfer entropy for a single lag $k$ (equation (2.9)), in terms of four separate joint entropy terms. Following equation (2.4), these may be estimated from the data using a non-parametric estimation of the probability density functions. For multivariate normal statistics, equations (2.9) and (2.3) coincide [13].

# 3. Methods

We calculate linear transfer entropy using ordinary least-squares regression, by comparing the variance of the residuals in the joint vector space $\{Y_t, Y_{t-k}, X_{t-k}\}$ against those in the independent vector space $\{Y_t, Y_{t-k}\}$, following equation (2.3).

To detect nonlinear transfer entropy, we perform non-parametric estimation of the probability density functions to calculate the joint entropy terms in equations (2.5) and (2.7). The density is estimated using a multidimensional histogram approach. Histograms have been shown to be effective in estimating entropy, [18,23], but there is a trade-off between bias and variance when partitioning the sample space. The choice of bins impacts the value of the calculated transfer entropy; too fine a partition introduces bias under finite sample size. In this paper we partition the sample space using marginal equiquantization, which treats each dimension independently in generating quantile bins such that each marginal bin contains roughly equal numbers of data points [24]. These are used to construct multidimensional histograms for estimating the probability distribution. Interestingly, to compute relative entropy measures such as mutual information or transfer entropy with a histogram approach, it is sufficient to compute the relative frequencies in each bin, even when bins have non-equal sizes. A note in appendix A explains why.

The equiquantization method has been used previously for the calculation of transfer entropy [18,24,25] and it has been shown that it can improve the performance under small sample sizes, by presenting a more precise representation of asymmetric or non-smooth distributions. We observe that using marginal quantile bins can be more robust to varying coarseness than equal-sized bins, as large gradients in the probability distribution function can be captured more closely without the introduction of additional information through refining the partition.

In estimating Shannon entropy, the coarseness of the partition directly impacts the numerical value, with finely partitioned histograms returning larger entropy values over the same data, since more information is introduced. This effect should cancel out in the calculation of transfer entropy; however, we observe instead that more bins generally results in larger transfer entropies for the same data, which amplifies both signal and noise. We therefore adopt a parsimonious approach in this paper, using a small number of bins compatible with a sufficient resolution, to capture the information transfer. We tested granular partitions using between three and eight bins per dimension, finding comparable results in each case. We report the results using histograms of six bins per dimension, a partition size which leads to good and meaningful results for each of the currencies analysed.

It is a feature of the non-parametric estimation of entropy that the absolute scale of the transfer entropy measure has only limited meaning. To detect causality, a relative position must be considered. A simple technique proposed by Marschinski & Kantz [18] is the effective transfer entropy (ETE), derived by subtracting from the observed transfer entropy an average transfer entropy figure calculated over independently shuffled time series, which destroys the temporal order and hence, in expectation, the time-dependence required for there to be any statistical causality.

We adopt a shuffling approach producing 100 null-hypothesis transfer entropy values from independently shuffled time-series over the same domain, containing no causal relationships. By calculating the mean and standard deviation of the shuffled transfer entropy figures, we estimate the significance of a causal result as the distance between the result and the average shuffled result, standardizing by the shuffled standard deviation

$$Z := \frac{\mathrm{TE} - \bar{\mathrm{TE}}_{\mathrm{shuffle}}}{\sigma_{\mathrm{shuffle}}}, \tag{3.1}$$

where TE is the transfer entropy of the temporally ordered sample, $\bar{\mathrm{TE}}_{\mathrm{shuffle}}$ is the average transfer entropy over all shuffled realizations and $\sigma_{\mathrm{shuffle}}$ is the standard deviation of the sample of shuffled realizations.

This quantity corresponds to the degree to which the result lies in the right tail of the distribution of the zero-causality shuffled samples, and hence how unlikely the result is due to chance. Therefore, the $Z$-score figure represents the significance of the excess transfer entropy in the unshuffled case. We compute the $Z$-score in equation (3.1) for both linear and nonlinear results.

To justify the usage of these techniques in detecting causal relationships in practice, we first validate the methodology using coupled time series of predefined causative relationships.

## 3.1. Synthetic geometric Brownian motion

We validate the approach by generating synthetic data following a directionally coupled random walk. First, we generate a driving series, following a discrete geometric Brownian motion (GBM)

$$X_{t+1} = (1 + \mu)X_t + \sigma X_t \, \eta_t, \tag{3.2}$$

where $\eta_t$ is a normally distributed random noise $\eta_t \sim \mathcal{N}(0, 1)$, and $\mu$ and $\sigma$ are respectively drift and diffusion coefficients. Then we produce a second series $Y_t$, which augments a linear dependence on the driving series $X$ with second linear dependence on an independent GBM process $X'$. The dependence in both cases is defined at a predetermined lag length $L$, and the strength of these dependencies is determined by some coupling constant $\alpha$, such that with $\alpha = 0$, $Y_t \equiv X'_{t-L}$ i.e. $Y$ is wholly independent of $X$, and in the limiting case $\alpha = 1$, $Y$ precisely follows the driving series $X$.

$$Y_t = \alpha X_{t-L} + (1 - \alpha)X'_{t-L}. \tag{3.3}$$

## 3.2. Synthetic coupled logistic map

We generate nonlinear coupled time series using a coupled logistic map. This system can be represented in terms of two stationary difference equations; the independent series is defined by the difference equation given by the general update function $f(X)$

$$f(X_t) = X_{t+1} = rX_t(1 - X_t), \tag{3.4}$$

where $X_t$ is the value of $X$ at time $t$, and $r$ is a parameter which in fact defines the dynamical state of the system. Following Hahs & Pethel [26], we take $r = 4$ so the function evolves chaotically. We then introduce a second map, which is dependent on the first, taking the form

$$Y_t = (1 - \alpha)rY_{t-L}(1 - Y_{t-L}) + \alpha g(X_{t-L}), \tag{3.5}$$

where $\alpha \in [0, 1]$ is the cross-similarity, or coupling strength, and $g(x)$ is a coupling function which may be chosen to produce different dynamic effects. In this case only $L = 1$ can be used. We follow the choice of Boba *et al.* [23] and Hahs & Pethel [26] for the coupling function

$$g(X_t) = (1 - \epsilon)f(X_t) + \epsilon f(f(X_t)), \tag{3.6}$$

where $\epsilon \in [0, 1]$ represents the coupling strength, describing the extent to which $Y_{t+1}$ depends on $f(f(X_t))$. It should be noted that the logistic map, in contrast to geometric Brownian motion, is a deterministic, albeit chaotic system, and that therefore $f(f(X_t))$ is equivalent to $X_{t+2}$. The extent of this anticipatory effect is driven by the selection of the $\epsilon$ parameter. We follow Hahs & Pethel in selecting $\epsilon = 0.4$. Indeed, as $\alpha$ increases, with large $\epsilon$, the direction of information transfer is less clear, as $Y_t$ contains more information about the future values of $X$.

# 4. Validation with synthetic data

In order to validate the autoregressive and information-theoretic approaches to detecting statistical causality, we apply these to the calculation of transfer entropy for synthetic data generated by both linear and nonlinear coupled time series, of increasing coupling strength. The computations are performed over the relative returns of the synthetic signals generated as described in §§3.1 and 3.2. It is important to note that many areas for which such statistical studies are important are constrained by limited available data. Therefore, for a technique of causal inference to be useful, it must perform even with small sample sizes. In §5 we apply both classical and information-theoretic methodologies to financial data with sample sizes of the order of $10^4$, and so we first evaluate the performance using synthetic time series of 15 000 observations, with known causal coupling, before confirming that the

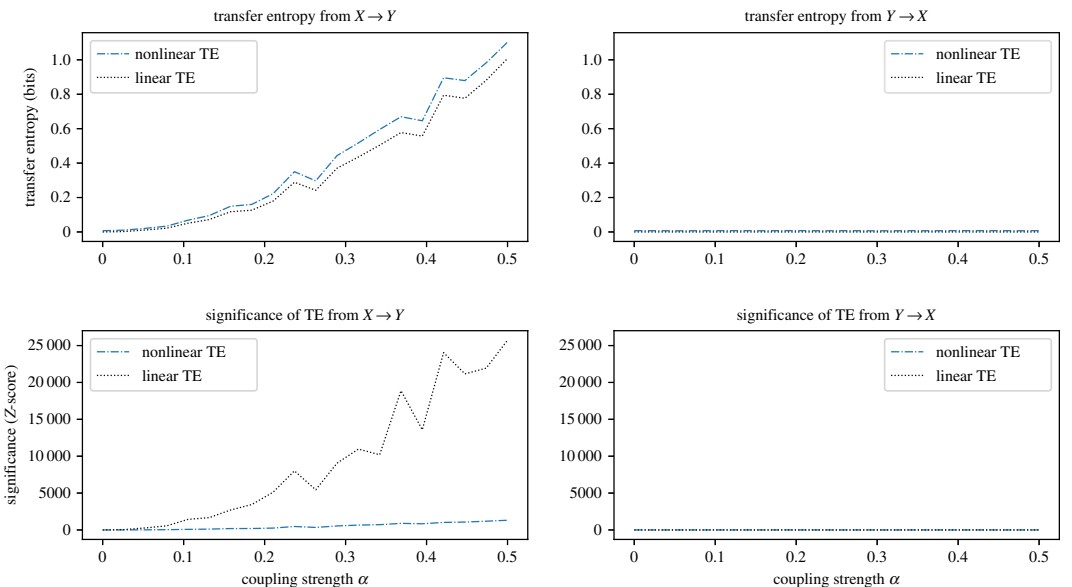

**Figure 1.** Demonstration that both linear and nonlinear transfer entropy methods detect statistical causality for linearly coupled synthetic data. The plots are calculated from the mean over 10 realizations using 15 000 data points of the synthetic random walk process from equations (3.2) and (3.3) with $\mathbb{E}[\mu] = 0.1$ and $\mathbb{E}[\mu'] = 0.1$, each with standard deviations of 0.1, and $\mathbb{E}[\sigma] = 0.025$ and $\mathbb{E}[\sigma'] = 0.025$, each with standard deviations of 0.01. Nonlinear transfer entropy is calculated using a quantile histogram of six classes per dimension. The Z-score of each result is also plotted for both methods. We observe no significant transfer entropy in the non-causal direction $Y \to X$. For the direction $X \to Y$, the Z-score of the linear technique is greater by orders of magnitude than the information-theoretic technique. However, both techniques correctly detect the linear transfer entropy, presenting largely equivalent results.

technique can detect the same causal coupling with as few as 500 samples. The results of this evaluation for increasing signal strength, using sample sizes of 500, 1000, 2500, 10 000 and 100 000, are presented in table 2 in appendix A.

## 4.1. Linear process causality validation

We calculate the directional information transfer from the driving series to the dependent series, and in the reverse direction, using both autoregressive and information-theoretic approaches for the linearly coupled system of GBM walks defined by equations (3.2) and (3.3). Parameters of the independent time series $\mu$ and $\sigma$ are generated independently for each realization and are distributed normally with $\mathbb{E}[\mu] = 0.1$ and $\mathbb{E}[\sigma] = 0.025$, and standard deviations of 0.1 and 0.01 respectively. Parameters of the endogenous series are also distributed normally around $\mathbb{E}[\mu'] = 0.1$ and $\mathbb{E}[\sigma'] = 0.025$, with standard deviations of 0.1 and 0.01. Figure 1 shows the results for coupling strengths from $\alpha = 0.0$ to $\alpha = 0.5$. For each coupling strength, a dataset is simulated over 15 000 time steps. Both techniques are applied to each dataset to calculate the information transfer, in both directions, with the results from $X \to Y$ and from $Y \to X$ plotted on separate axes.

In the information-theoretic approach we calculate transfer entropy using histograms with quantile binning of six classes per dimension. We generate multiple synthetic coupled random walks, calculating transfer entropy and Z-scores for each realization, and reporting the mean values. Quantile bins are generated independently for each realization. We validate causality using the significance testing approach described in §3.

As can be observed from figure 1, the qualitative correspondence between both methods is clearly visible, and quantitatively the results are similar. Additionally, the one-way direction of information transfer is accurately detected, with large transfer entropy and Z-scores observed in the direction of $X \to Y$, and approximately zero values in the opposite direction.

## 4.2. Nonlinear process causality validation

We calculate the directional information transfer from the driving series to the dependent series, and in the reverse direction, using both autoregressive and information-theoretic approaches for the nonlinear coupled logistic map system from equations (3.4), (3.5) and (3.6).

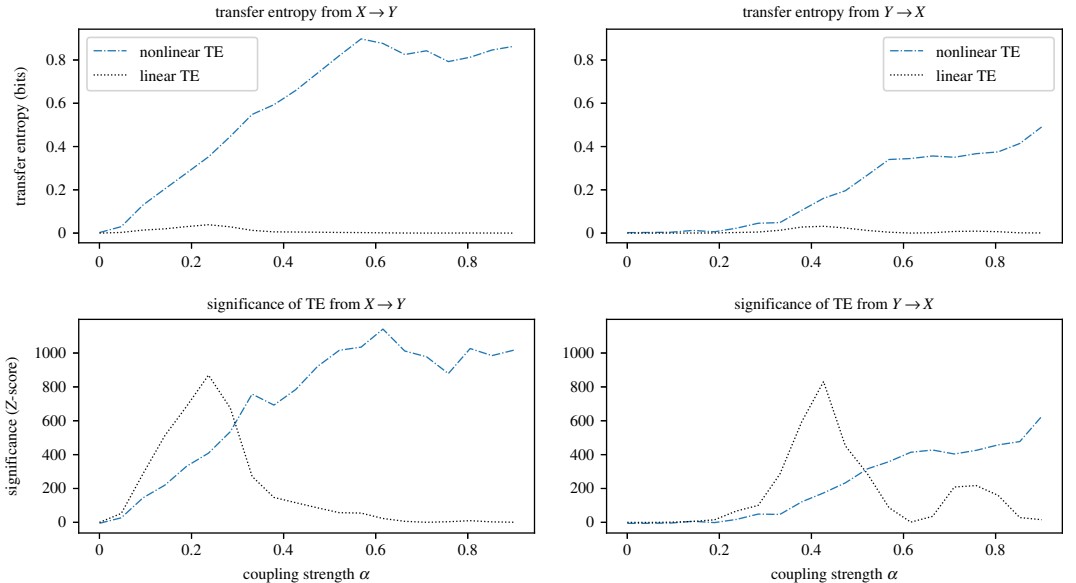

**Figure 2.** Demonstration that the nonlinear causal relationship in synthetic data generated from equations (3.5) and (3.6) is correctly detected only by the nonlinear method. The plots are calculated from the mean over 10 realizations over 15 000 data points of the synthetic coupled logistic map process, with $\epsilon = 0.4$. Nonlinear transfer entropy is calculated using a histogram of six classes per dimension, partitioned using marginal equiquantization. The Z-score for each result is also plotted for both methods. In the direction $X \rightarrow Y$, the classical approach is unable to detect the known dependency of Y on X, while detecting small spurious signals. By contrast the information-theoretic approach correctly identifies the relationship. We note that for $\alpha > 0.2$ the information-theoretic method detects information transfer in the other direction. This is observed to increase as $\alpha$ approaches 1, and corresponds to an expected nonlinear anticipatory signal.

In the information-theoretic approach we calculate transfer entropy again using histograms with quantile binning of six classes per dimension, generating bins independently for each realization.

Figure 2 shows the mean transfer entropy results for 15 000 synthetic data points. We observe that, for this system, the linear method is incapable of detecting the causal relationship; it finds some significant information transfer (at a near-zero level, which we interpret as the linear component), but fails to represent the expected exposure–response relationship and does not correctly identify the expected anticipatory signal in the reverse direction. The information-theoretic method, by contrast, produces results which better represent the increasing coupling strength relationship, and direction of Granger causality in the system. The technique also detects Granger causality from $Y$ to $X$, for large values of $\alpha$. This effect was also observed by Liang [15] and Hahs & Pethel [26], and corresponds to the anticipatory nature of the coupling. Specifically the coupling function $g(x)$ involves repeated application of the update function $f(x)$ from equation (3.4), from which it can be seen that $f(f(X_t))$ is equivalent to $X_{t+2}$ so, for large $\alpha$, $Y_t$ will contain increasing amounts of the future information of $X_t$. In fact, at large coupling strengths approaching $\alpha = 1$, the observed transfer entropy from $X$ to $Y$ begins to decrease, as more information exists in $Y$ about its future evolution.

The results of these validation experiments suggest that the information-theoretic approach is superior in detecting causal signals, being model-free and so able to detect relationships of more complex, nonlinear modes.

## 4.3. Decay of causal signals with lag length

As a final validation exercise, we explore the performance of the methods in detecting signals in coupled time series when the lag of the relationship is unknown. In general, it is expected that causal links should be strongest at time-lags closest to the true signal lag, and gradually decay as the time-lag considered is increased. However, the complexity of causative relationships, particularly where any feedback exists between the time series, suggests that there could also be multi-modal causalities, operating at different lags.

We use the coupled GBM system defined in equations (3.2) and (3.3) to create a coupling of a fixed lag $L = 6$, and then perform both autoregressive and information-theoretic analysis to detect the transfer

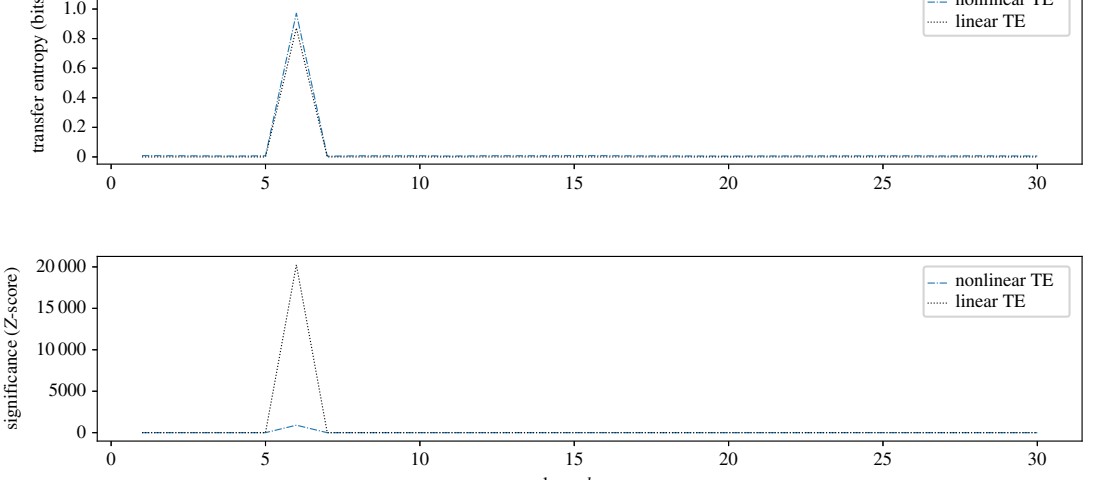

**Figure 3.** Demonstration that both methods identify the true lag $L = 6$ with maximal transfer entropy. Nonlinear transfer entropy is calculated using a quantile-binned histogram, of six classes per dimension, over 15 000 points. Each result is calculated from the mean over 10 realizations. In the $\alpha = 0.5$ case we observe a non-zero transfer entropy at the search length $k = L$, which corresponds to the known lag of the causal coupling $L = 6$. No significant transfer entropy is observed by either technique, at all other lags.

entropy at time-lags from $k = 1$ up to $k = 30$. The information-theoretic approach is applied using histograms partitioned into six classes per dimension, using marginal equiquantization, and is shown to identify the specific temporal lag between driving and dependent series. The results are shown in figure 3.

We observe the identification of the correct lag of the causal signal, which is marked by peaks at lag equal to the true lag $L = 6$ of the causal relationship, and zero elsewhere.

# 5. Results with real data

Having confirmed that the information-theoretic approach is able to detect both linear and nonlinear causalities, we apply the technique to investigate the effect of changes in social media sentiment on cryptocurrency returns, and vice versa. We also apply the linear method, and compare estimates from both techniques to establish whether linear or nonlinear dynamics dominate the observed causal relationship.

We estimate information transfer over 24-month windows, rolling forward with a stride of one month from the earliest market data available to September 2018. Each window contains 17 545 observations of price and sentiment, sampled at hourly intervals. Price is taken as the combined close price, on the hour, over an aggregation of exchanges (see appendix A.3). Social sentiment is estimated from NLP analysis of Twitter tweets and StockTwits during the preceding hour; we quantify this sentiment as the sum of positive messages in the previous hour. In early periods of the data, infrequently some hours have no messages; in these cases we forward-fill from the previous hour, making the assumption that sentiment does not drop to neutral in these cases. To handle non-stationarity in the data, we take the difference between the logarithms of the values at times $t$ and $t - 1$. This differencing is applied to both time series.

The choice of timescale in aggregating raw sentiment data involves a trade-off: with too fine a timescale, there are not enough messages to estimate sentiment, but a long timescale represents a low-resolution sample which loses too much information about the underlying time series. Since cryptocurrencies are traded in real time on electronic exchanges, we hypothesize that causal signals between sentiment and returns operate at sub-hourly timescales; hourly aggregation is the smallest time period available in the data, and so this aggregation of sentiment is used.

For the information-theoretic approach, it is observed that performing the analysis with histograms of equal-width bins gives different results depending on the number of bins selected. Specifically, partitioning the axes of the sample space into odd-numbers of bins produces no significant result over this data, suggesting the information is captured mostly from the middle peak of the distribution.

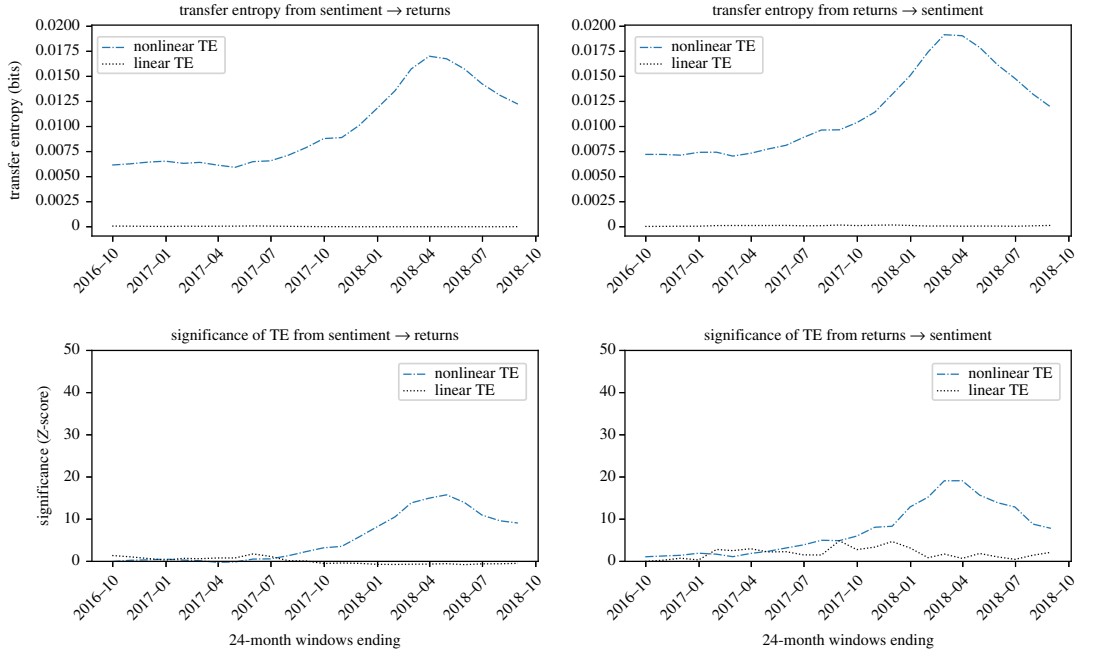

**Figure 4.** Evidence that BTC sentiment and returns are causally coupled in both directions in a nonlinear way. Nonlinear TE is calculated by multidimensional histograms with six quantile bins per dimension. Z-scores, calculated over 100 shuffles, show a high level of significance, especially during 2017 and 2018, in both directions, with the peak significance $Z = 21.3$ observed in the direction of returns to sentiment in the 24-month window ending February 2018.

However, we note that the use of quantile binning avoids this issue, finding both odd-numbered and even-numbered bin counts to provide similar results, suggesting a key benefit in using quantile bins for the calculation of transfer entropy. Accordingly, in this analysis we partition the sample space into quantile bins, using six classes per dimension, having validated this choice in §4. The histogram bins for the nonlinear approach are calculated once, using the full dataset for each currency, and then they are applied across all windows. In selecting an appropriate partition, further bias is inevitably introduced. By calculating appropriate bins for each window, the results cannot be directly compared between windows. However, the growth in message volumes over time means that selecting bins sized to capture the full spread of values would also introduce a bias, since such bins are more suited to the later months than the earlier months. Since the granularity of the histogram partition also impacts the transfer entropy value, we perform significance tests over each window independently, calculating the Z-scores and comparing these across windows and currencies. We report the windows with greatest significance using a time-lag of $k = 1$ hour. Performing the analysis using longer time-lags shows weaker causal signals over this data.

Plots showing the information transfer for the four cryptocurrencies investigated are reported in figures 4–7. Z-scores for each result are also presented; for comparison across figures, Z-score axes use the same scale.

For BTC, in figure 4, we detect a strong causative signal, of roughly similar scale in both directions of sentiment to BTC returns and in the reverse direction. Analysis of the Z-scores confirms the significance in both directions, with the net information transfer greater in the direction of returns to sentiment, with mean net transfer entropy of −0.0016 bits.

LTC, in figure 5, shows a similar pattern to BTC, although in this case the predominant direction of information transfer is reversed, with mean net transfer entropy of 0.0039 bits. The Z-scores show the significance of sentiment to returns is consistently greater than in the reverse direction, and also stronger than the other currencies. In addition, there is notably a large linear component to the information transfer, which is unique to LTC.

XRP, in figure 6, shows a clear nonlinear causality from sentiment to returns, with a significant but smaller causal relationship also in the opposite direction. The mean net transfer entropy is measured for this period as 0.0022 bits. The signal is more significant from sentiment to returns, and especially in the periods ending in 2018.

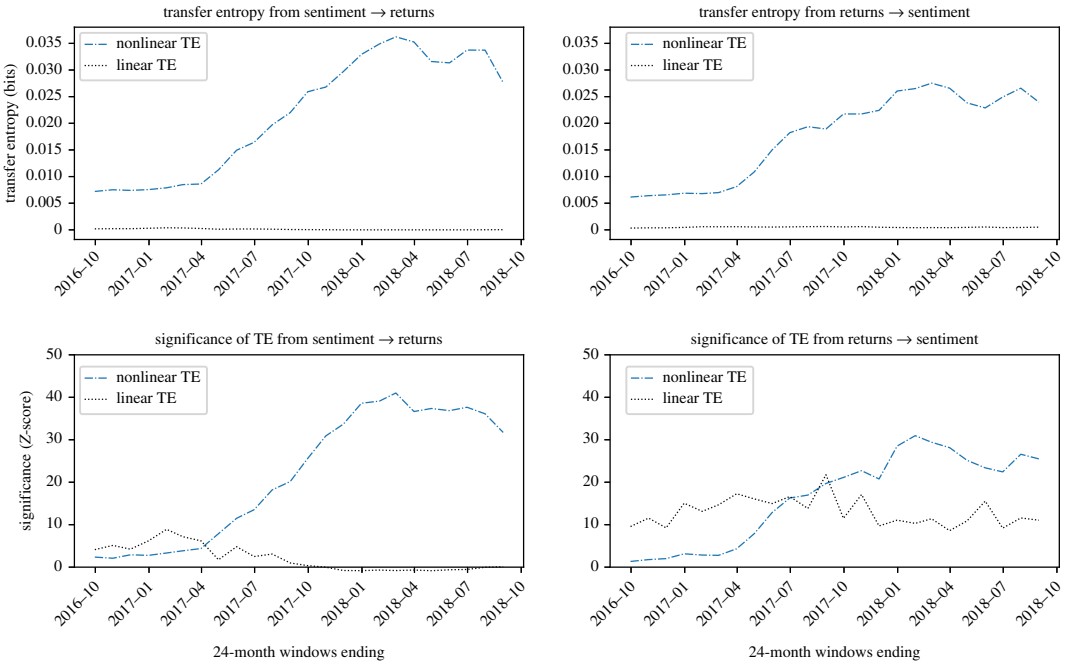

**Figure 5.** Evidence that LTC returns and sentiment are causally coupled in both directions in a nonlinear way, with sentiment having a larger influence on returns than the other way round. Nonlinear TE is calculated by multidimensional histograms with six quantile bins per dimension. $Z$-scores, calculated over 100 shuffles, show a significant signal in both directions, with the net information transfer generally operating in the direction of sentiment to returns. Peak significance of $Z = 42.8$ is observed from sentiment to returns in the 24-month window ending February 2018

ETH, in figure 7, shows an interesting and unique behaviour. In particular, there appears to be, initially, a significant signal which collapses in the windows ending around January 2018. The information transfer at its initial peak is greater from returns to sentiment; however, over the whole period the effect is more bidirectional, with mean net transfer entropy of zero. This suggests a slight phase change, with sentiment driving returns after the collapse of the signal, although no net significance is observed after this point in either direction. This strongly indicates another driving mechanism, the effect of which first becomes present around January 2016 (due to 24-month windows). This effect is likely to be associated with the rapid price movements at the time.

## 6. Conclusion

Information-theoretic and autoregressive techniques were developed and validated on coupled random walks and chaotic logistic maps, confirming the ability of both techniques to detect linear information transfer, and of the information-theoretic technique to detect nonlinear information transfer. Following validation, the techniques were applied to historical data describing social media sentiment and cryptocurrency prices to detect information transfer between changes in sentiment and price returns.

The information-theoretic investigation detected a significant nonlinear causal relationship in BTC, LTC and XRP, over multiple timescales and in both directions of sentiment to returns and returns to sentiment. The effect was strongest and most consistent for LTC and XRP, and in both cases the net information transfer was in the direction of sentiment to returns. ETH and BTC were observed to show significant net information transfer in the direction returns to sentiment, although separate dynamics were observed in ETH which see a collapse in significance.

The significance tests confirm the existence of causally coupled relationships, though the strength of these relationships are challenging to accurately quantify from the data, especially for the purposes of comparison between different time series, and between the linear and nonlinear results over the same data. However, the significance values themselves offer the possibility of quantifying the strength of causality, which may be used as a proxy when using transfer entropy as a tool for detecting statistical causality. With this work we demonstrate that the dynamics of the causative relationship is nonlinear, as the autoregressive technique was able to detect only small or zero causality in either direction, for any of the currencies. The mean $Z$-scores of results observed by linear and information-theoretic

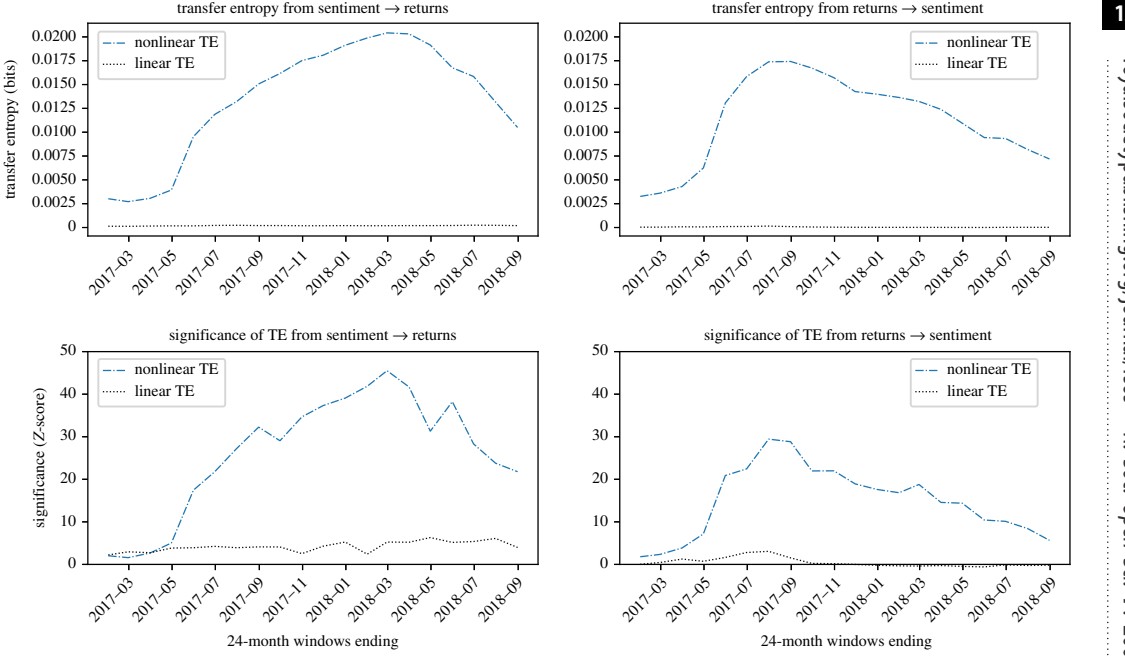

**Figure 6.** Evidence that XRP returns and sentiment are causally coupled in both directions in a nonlinear way, with the prevailing direction of information transfer flowing from sentiment to returns in the first period, and from returns to sentiment in the second. Nonlinear TE is calculated by multidimensional histograms with six quantile bins per dimension. Z-scores, calculated over 100 shuffles, show a small but clear significant signal, in both directions, which decays rapidly towards January 2018 and does not recover afterward. Peak significance of $Z = 41.9$ is observed from sentiment to returns in the 24-month window ending January 2018.

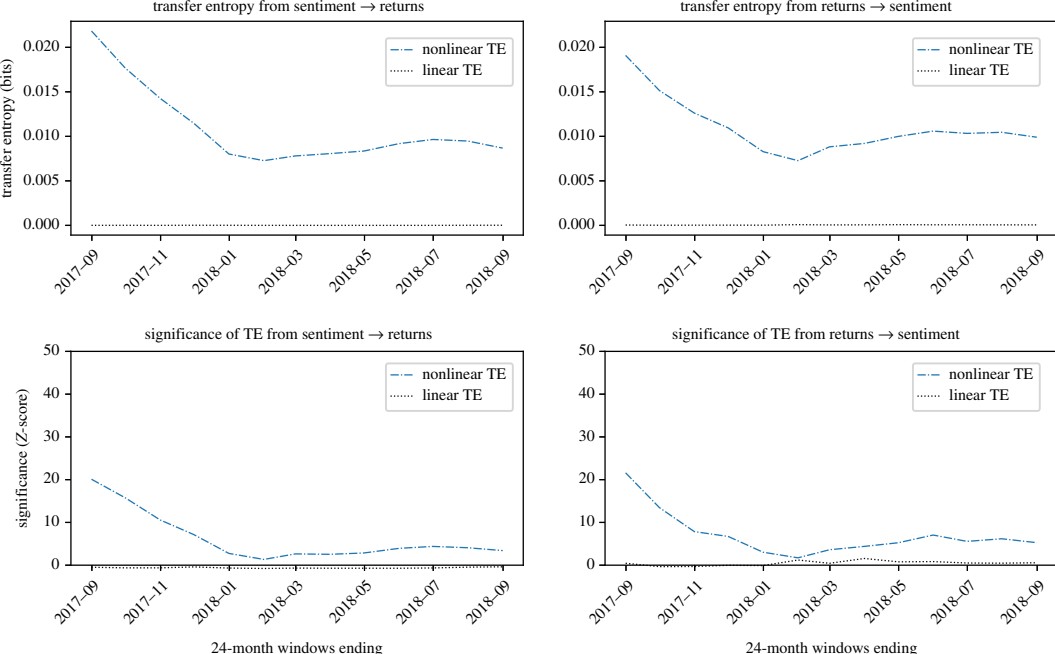

**Figure 7.** Evidence that ETH returns and sentiment are causally coupled in both directions in a nonlinear way. Overall this coupling is of lower significance compared to the other currencies investigated. Nonlinear TE is calculated by multidimensional histograms with six quantile bins per dimension. Z-scores, calculated over 100 shuffles, indicate initial significance for windows ending before 2018, followed by low significance after the collapse in signal strength, which must therefore begin around January 2016. Peak significance is observed in the first window of the study, with $Z = 20.6$ in the direction of sentiment to returns, and $Z = 19.9$ in the direction of returns to sentiment.

methods are presented for each currency in table 4. Our results show that histograms of marginal equiquantization, in concert with permutation tests, can be used in practice to detect nonlinear information transfer between time series, even where classical statistical techniques fail. Let us point out that there is a risk of assuming ergodicity in the results; we have shown that the level of causation varies historically within the sample time period, and there is no evidence that the observed causal relationships will continue unchanged.

Ethics. In this manuscript, there are no data or topics that need ethical approval.
Data accessibility. The source code and data can be found at https://github.com/ZacKeskin/. Synthetic data can be generated using the functions in the PyCausality package within test utils, in Time_Series_Generate.py.
Competing interests. There are no competing interests.
Authors' contributions. Z.K. wrote the code, and drafted the first version of the paper. T.A. directed the research proposing the methodologies and applications. Both authors contributed equally to the final draft of the paper.
Acknowledgements. The authors acknowledge Thársis Souza in advising on the method of testing for linear Granger causality, with thanks along with Yuqing Long, whose data collation and wrangling was a great help. Finally, we thank PsychSignal for providing their market sentiment data for this academic study.
Funding. T.A. acknowledges support from ESRC (ES/K002309/1), EPSRC (EP/P031730/1) and EC (H2020-ICT-2018-2 Fin-Tech 825215).

# Appendix A

## A.1. Estimates of mutual information and transfer entropy from relative frequencies

In order to compute the mutual information or the transfer entropy, we must estimate entropies from equation (2.4). The histogram approach discretizes the continuous sample space by partitioning it into bins that—in general—are not assumed to be of equal size, and so transforms the integral into a sum.

$$H(X) = -\int_{-\infty}^{+\infty} p(x) \log p(x)\, dx \simeq -\sum_{x_k} \hat{p}(x_k) \log(\hat{p}(x_k)) v(x_k), \tag{A 1}$$

where $\hat{p}(x_k)$ is the estimate of the probability density function $p(x)$ computed for the bin $x_k$ that contains the observation $x$. The quantity $v(x_k)$ is the proportion of the support of the probability density function which is occupied by the bin. This generalizes to multiple dimensions. The estimate of the probability density function $\hat{p}(x_k)$ can be computed from the number of observations $n(x_k)$ that fall in each bin via

$$\hat{p}(x_k) = \frac{n(x_k)}{N v(x_k)}, \tag{A 2}$$

where $N = \sum_{x_k} n(x_k)$ is the total number of observations. The entropy estimation is therefore

$$H(X) \simeq -\sum_{x_k} \frac{n(x_k)}{N} \log\left(\frac{n(x_k)}{N v(x_k)}\right). \tag{A 3}$$

This quantity depends on the relative volumes of the bins, and hence do the estimates for $H(Y)$ and $H(X, Y)$ also require these to be calculated. However, in the mutual information $I(X; Y) = H(X) + H(Y) - H(X, Y)$ the terms containing the volumes cancel, and its expression turns out to be dependent only on the relative frequencies

$$
\begin{aligned}
I(X; Y) \simeq &\sum_{x_k, y_j} \frac{n(x_k, y_j)}{N} \log\left(\frac{n(x_k, y_j)}{N}\right) \\
&- \sum_{x_k} \frac{n(x_k)}{N} \log\left(\frac{n(x_k)}{N}\right) - \sum_{y_j} \frac{n(j_j)}{N} \log\left(\frac{n(y_j)}{N}\right).
\end{aligned}
\tag{A 4}
$$

The same applies to the transfer entropy. This is a remarkable simplification of the computation of these quantities, which shows how they can be estimated from histograms simply in terms of the relative frequencies, for any kind of binning partition.

## A.2. Source code

All analysis for this paper was performed using a Python package (PyCausality) created by the lead author. This is maintained on the author's public GitHub profile, which can be found at https://github.com/ZacKeskin/PyCausality. For the latest release this can be simply installed via PyPi using pip.

Synthetic data as presented in this paper can be generated using the functions made available in the package test utils, in Time_Series_Generate.py. This contains code to produce coupled time series either of the chaotic logistic map or of geometric Brownian motion. To replicate the validation experiments in §4, time series should be generated with parameters $T = 1$, $N = 15\,000$ for the coupled random walk, and with parameters $T = 10$, $N = 15\,000$ and $\epsilon = 0.4$ for the coupled logistic map.

Ongoing maintenance and pre-release development of the package will be made available through this repository, and contributors may fork code and submit pull requests to develop this further.

## A.3. Data

The social sentiment data were provided courtesy of PsychSignal, and may be made available pending request to the authors. The data take the form of the number of positive messages and the number of negative messages, publicly shared on either Twitter or StockTwits, associated each hour with the cryptocurrencies in question. The association is detected via the use of a 'hashtag' (or 'cashtag') which takes the form of #BTC or #Bitcoin (for example) on twitter, or $BTC on StockTwits. For inclusion in the dataset, the message must contain one of the tags described in table 1.

**Table 1.** Hashtags used to map social media messages to specific cryptocurrencies.

| curency | tag | currency | tag |
| --- | --- | --- | --- |
| bitcoin | BTC | litecoin | LTC.X |
| bitcoin | BCOIN | litecoin | LTCUSD |
| bitcoin | BTC.X | ripple | XRP.X |
| bitcoin | BTCEUR | ripple | XRPBTC |
| bitcoin | BTCGBP | ripple | XRPUSD |
| bitcoin | BTCUSD | ethereum | ETH |
| bitcoin | GBTC | ethereum | ETH.X |
| bitcoin | SGDBTC | ethereum | ETHUSD |

Price data is the hourly close in USD, obtained via CryptoCompare's public API. This provides a combined average over multiple exchanges, where prices are available. For further details, the documentation is available at https://min-api.cryptocompare.com/.

## A.4. Tables

Analysis was undertaken to validate the performance of linear and nonlinear techniques over multiple parameter values for sample size and signal strength. Table 2 contains the Z-scores of transfer entropy

**Table 2.** Z-scores of transfer entropy from $X$ to $Y$, for coupled GBM system described in §4. Calculated using the information-theoretic technique for increasing sample size $n$ and coupling strength $\alpha$.

| n | 0 | 0.1 | 0.3 | 0.5 |
| --- | --- | --- | --- | --- |
| 500.0 | −0.0 | 0.8 | 6.5 | 9.8 |
| 1000.0 | 0.1 | 10.7 | 13.5 | 42.1 |
| 2500.0 | 0.2 | 6.5 | 69.9 | 57.1 |
| 10000.0 | −0.2 | 57.6 | 272.5 | 225.3 |
| 100000.0 | −0.1 | 1339.7 | 2387.3 | 2254.8 |

**Table 3.** Net information transfer statistics in units of bits determined over all windows of the study for each cryptocurrency. Mean values show that in time, more information was transferred from sentiment to price for LTC and XRP, with more information transfer in the direction of price to sentiment for BTC and ETH. It should be noted that due to the overlapping strides of the windows, periods during the middle of the sample are weighted more heavily; however, the figures appropriately present the relative direction of information transfer over the 48-month period of the study, so are presented for the sake of comparison. Significance scores are presented in table 4.

| | mean | 10th percentile | 90th percentile |
|---|---|---|---|
| BTC | −0.0016 | −0.0032 | −0.0005 |
| LTC | 0.0039 | 0.0004 | 0.0086 |
| XRP | 0.0022 | −0.0036 | 0.0074 |
| ETH | −0.0001 | −0.0014 | 0.0023 |

**Table 4.** Table of average Z-scores across all windows of observation for each currency. We observe nonlinear values generally an order of magnitude greater than the linear values, with the exception of LTC for which we see relatively large linear components. However, the nonlinear transfer entropy remains considerably larger than the linear values, even for LTC.

| | (returns → sentiment) | | (sentiment → returns) | |
|---|---|---|---|---|
| | linear | nonlinear | linear | nonlinear |
| BTC | 0.1 | 5.1 | 2.0 | 7.6 |
| LTC | 2.2 | 22.4 | 11.8 | 17.3 |
| XRP | 4.1 | 24.1 | 0.4 | 14.4 |
| ETH | −0.6 | 6.5 | 0.5 | 6.8 |

calculations using marginal equiquantization, for coupled random walks of increasing coupling strength and sample size.

Table 3 contains results quantifying the net information transfer across windows of the study for each cryptocurrency. Net information transfer is calculated as $TE_{X \to Y}^{(1)} - TE_{Y \to X}^{(1)}$, and is given in units of bits. Table 4 contains the average Z-scores across all 24-month windows, in both directions, for each currency. These are calculated for both linear and nonlinear techniques.

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
