## [Reviewer comments · Royal Society Open Science]

Review History

RSOS-200863.R0 (Original submission)

Review form: Reviewer 1 (Hermann Elendner)

Is the manuscript scientifically sound in its present form?

Yes

Are the interpretations and conclusions justified by the results?

Yes

Is the language acceptable?

Yes

Do you have any ethical concerns with this paper?

No

Have you any concerns about statistical analyses in this paper?

No

Recommendation?

Accept with minor revision (please list in comments)

Comments to the Author(s)

The paper is an original exercise in employing the information-theoretic concept of entropy as a means to study the flow of economic information in cryptocurrency markets. This allows to extend common studies of Granger causality to non-linear dependencies. The application on sentiment data and cryptocurrency prices is meant to show that the nonlinearity is necessary to capture the information flow.

First some small points that absolutely need fixing:

- You repeat in many places that sentiment causes /price/. This would be both statistically (non-stationarity) as well as economically (arbitrage) extremely worrying. Now, you do what should be done (take log differences, and say so in §2 of Sec 5), but the terms should be straightened out in the entire paper: you study the effects of changes in sentiment on returns! (and vice versa)

- The "vice versa" is natural in the context of Granger causality and transfer entropy, but your liberal use of "causality" as synonymous with those will likely offend many theorists. It would certainly have offended Hume, your first reference. All this can still be fine if you declare this "equivalence" for the scope of your paper; however, when you (implicitly) dismiss other meanings of causality (including the standard and intuitive one), this goes too far: shuffling will destroy, in expectation, the time dependence you measure -- and thus I have no objections to your method -- but it will certainly not "hence [destroy] any causality" (p 6; just measure mass and volume of a liter of water every day under identical circumstances to conclude that there is no causal relationship between those 2 from the 2 constant lines observed)

- You also over-reach in your statement on p 11 claiming to provide evidence of sub-hourly dependence patterns. You have indication of a stronger pattern as short as one-hour lags; there is no reason to conclude about frequencies you cannot study.

Less technical points:

- While you link well to the literature on sentiment analysis in crypto markets, you do not mention at all that your problem is at the heart of a large financial literature on price discovery.

- This would have also led you to the obvious question, do you find "causality" between returns of different coins?

- The paper would also profit from at least some discussion of why you believe the standard approach to handle non-linearities is insufficient, namely transforming the driver process.

- In relation to this, the fact that you find mostly no linear but non-linear dependence begs the question how the relation does look like. You should be able to estimate the relationship, ideally provide a graph.

- This is not a purely statistical point, actually the standard pricing models in financial economics will explain returns in factor models where the factors are non-linear functions, potentially of past prices! (momentum)

- The other elephant in the room which goes unmentioned is liquidity. You should have trading volume data from cryptocoin and could thus address this easily with at least the Amihud measure.

- Consequently, this raises the point of alternative drivers. Given that you have done your placebo tests on generated data, why not show what causality you can detect from LTC sentiment changes on BTC returns, etc?

- Finally a marginal point which may be subjective, but I do not buy the argument that we should expect more effect of sentiment in crypto markets due to their limited role as media of exchange. Common stock is much harder to exchange, and many more unsophisticated investors are investing in stock markets. I do not claim that the speculative motive is smaller; I just don't believe your chain of argument to be convincing in this regard. And you do not even need it in the least: sentiment analysis is a strong (if controversial) field in financial economics outside and inside of cryptocurrency markets, your question is valid and important independent of your beliefs about how cryptos "should" be priced.

Review form: Reviewer 2

Is the manuscript scientifically sound in its present form?

Yes

Are the interpretations and conclusions justified by the results?

Yes

Is the language acceptable?

Yes

Do you have any ethical concerns with this paper?

No

Have you any concerns about statistical analyses in this paper?

No

Recommendation?

Accept with minor revision (please list in comments)

Comments to the Author(s)

The manuscript presents a study of non-linear causality detection between social sentiment and prices. The detection is done by using nonlinear transfer entropy.

The topics is timely and the authors develop it properly. The results are interesting for the interdisciplinary research community interested in quantitative modeling of financial market dynamics.

The paper can be published as it is, however the authors might consider to amend it according to the following minor comments:

1) The authors consider prices of four cryptocurrencies. In several other studies of casualties asset returns have been considered. Why they choose prices? Would a study done with returns produce the same or similar results?

2) the authors are using 24-month windows with a one month step. Most of the results of figures 4-7 therefore are highly correlated at successive time records. To better understand the dynamics of the nonlinear causality it would be interesting to see how the results change when the overlapping between time windows decreases;

3) the recent literature on sentiment signals extracted from tweets highlights the evergrowing role of bots in the dynamics of Tweeter. Did the authors perform any pre-processing to evaluate the presence or absence of bot actions in the construction of their sentiment data series? Are bot present? Do they have any role?

4) Trading of cryptocurrency and Twitter activity are geographically constrained (although markets for cryptocurrencies are active 24 hours a day). Are intraday patterns due to geographical activity of Asia, Europe, and Americas playing a role in the determination of transfer entropy?

5) Several of the references are appropriate and informative. However, other studies dealing with causality detection in financial markets are missing from the reference list. Authors might find a nice literature review in the paper Sandoval, L., 2014. Structure of a global network of

financial companies based on transfer entropy. *Entropy*, 16(8), pp.4443-4482 and select from it the literature they consider appropriate.

Decision letter (RSOS-200863.R0)

Dear Professor Aste

On behalf of the Editors, I am pleased to inform you that your Manuscript RSOS-200863 entitled "Information-theoretic measures for non-linear causality detection: application to social media sentiment and cryptocurrency prices" has been accepted for publication in Royal Society Open Science subject to minor revision in accordance with the referee suggestions. Please find the referees' comments at the end of this email.

The reviewers and handling editors have recommended publication, but also suggest some minor revisions to your manuscript. Therefore, I invite you to respond to the comments and revise your manuscript.

- Ethics statement

- Data accessibility

<http://datadryad.org/submit?journalID=RSOS&manu=RSOS-200863>

- Competing interests

- Authors' contributions

- Acknowledgements

- Funding statement

Because the schedule for publication is very tight, it is a condition of publication that you submit the revised version of your manuscript before 29-Jul-2020. Please note that the revision deadline will expire at 00.00am on this date. If you do not think you will be able to meet this date please let me know immediately.

- 1) A text file of the manuscript (tex, txt, rtf, docx or doc), references, tables (including captions) and figure captions. Do not upload a PDF as your "Main Document";
- 2) A separate electronic file of each figure (EPS or print-quality PDF preferred (either format should be produced directly from original creation package), or original software format);
- 3) Included a 100 word media summary of your paper when requested at submission. Please ensure you have entered correct contact details (email, institution and telephone) in your user account;
- 4) Included the raw data to support the claims made in your paper. You can either include your data as electronic supplementary material or upload to a repository and include the relevant doi

within your manuscript. Make sure it is clear in your data accessibility statement how the data can be accessed;

5) All supplementary materials accompanying an accepted article will be treated as in their final form. Note that the Royal Society will neither edit nor typeset supplementary material and it will be hosted as provided. Please ensure that the supplementary material includes the paper details where possible (authors, article title, journal name).

If your manuscript is newly submitted and subsequently accepted for publication, you will be asked to pay the article processing charge, unless you request a waiver and this is approved by Royal Society Publishing. You can find out more about the charges at <https://royalsocietypublishing.org/rsos/charges>. Should you have any queries, please contact openscience@royalsociety.org.

Kind regards,

Anita Kristiansen
Editorial Coordinator

on behalf of Professor Manfred Broy (Associate Editor) and Marta Kwiatkowska (Subject Editor)
openscience@royalsociety.org

Subject Editor Comments to Author (Professor Marta Kwiatkowska):

Comments to the Author:

Both reviewers (one recommended) appreciate the contribution and agree that the paper should be accepted with minor revisions. The reviews are quite detailed.
The code should be made available, as well as data (see comment from 1st reviewer).

Reviewer comments to Author:
Reviewer: 1

Comments to the Author(s)

The paper is an original exercise in employing the information-theoretic concept of entropy as a means to study the flow of economic information in cryptocurrency markets. This allows to

extend common studies of Granger causality to non-linear dependencies. The application on sentiment data and cryptocurrency prices is meant to show that the nonlinearity is necessary to capture the information flow.

First some small points that absolutely need fixing:

- You repeat in many places that sentiment causes /price/. This would be both statistically (non-stationarity) as well as economically (arbitrage) extremely worrying. Now, you do what should be done (take log differences, and say so in §2 of Sec 5), but the terms should be straightened out in the entire paper: you study the effects of changes in sentiment on returns! (and vice versa)

- The "vice versa" is natural in the context of Granger causality and transfer entropy, but your liberal use of "causality" as synonymous with those will likely offend many theorists. It would certainly have offended Hume, your first reference. All this can still be fine if you declare this "equivalence" for the scope of your paper; however, when you (implicitly) dismiss other meanings of causality (including the standard and intuitive one), this goes too far: shuffling will destroy, in expectation, the time dependence you measure -- and thus I have no objections to your method -- but it will certainly not "hence [destroy] any causality" (p 6; just measure mass and volume of a liter of water every day under identical circumstances to conclude that there is no causal relationship between those 2 from the 2 constant lines observed)

- You also over-reach in your statement on p 11 claiming to provide evidence of sub-hourly dependence patterns. You have indication of a stronger pattern as short as one-hour lags; there is no reason to conclude about frequencies you cannot study.

Less technical points:

- While you link well to the literature on sentiment analysis in crypto markets, you do not mention at all that your problem is at the heart of a large financial literature on price discovery.

- This would have also led you to the obvious question, do you find "causality" between returns of different coins?

- The paper would also profit from at least some discussion of why you believe the standard approach to handle non-linearities is insufficient, namely transforming the driver process.

- In relation to this, the fact that you find mostly no linear but non-linear dependence begs the question how the relation does look like. You should be able to estimate the relationship, ideally provide a graph.

- This is not a purely statistical point, actually the standard pricing models in financial economics will explain returns in factor models where the factors are non-linear functions, potentially of past prices! (momentum)

- The other elephant in the room which goes unmentioned is liquidity. You should have trading volume data from cryptocompare and could thus address this easily with at least the Amihud measure.

- Consequently, this raises the point of alternative drivers. Given that you have done your placebo tests on generated data, why not show what causality you can detect from LTC sentiment changes on BTC returns, etc?

- Finally a marginal point which may be subjective, but I do not buy the argument that we should expect more effect of sentiment in crypto markets due to their limited role as media of exchange. Common stock is much harder to exchange, and many more unsophisticated investors are investing in stock markets. I do not claim that the speculative motive is smaller; I just don't believe your chain of argument to be convincing in this regard. And you do not even need it in the least: sentiment analysis is a strong (if controversial) field in financial economics outside and inside of cryptocurrency markets, your question is valid and important independent of your beliefs about how cryptos "should" be priced.

Reviewer: 2

Comments to the Author(s)

The manuscript presents a study of non-linear causality detection between social sentiment and prices. The detection is done by using nonlinear transfer entropy.

The topics is timely and the authors develop it properly. The results are interesting for the interdisciplinary research community interested in quantitative modeling of financial market dynamics.

The paper can be published as it is, however the authors might consider to amend it according to the following minor comments:

1) The authors consider prices of four cryptocurrencies. In several other studies of casualties asset returns have been considered. Why they choose prices? Would a study done with returns produce the same or similar results?

2) the authors are using 24-month windows with a one month step. Most of the results of figures 4-7 therefore are highly correlated at successive time records. To better understand the dynamics of the nonlinear causality it would be interesting to see how the results change when the overlapping between time windows decreases;

3) the recent literature on sentiment signals extracted from tweets highlights the evergrowing role of bots in the dynamics of Tweeter. Did the authors perform any pre-processing to evaluate the presence or absence of bot actions in the construction of their sentiment data series? Are bot present? Do they have any role?

4) Trading of cryptocurrency and Twitter activity are geographically constrained (although markets for cryptocurrencies are active 24 hours a day). Are intraday patterns due to geographical activity of Asia, Europe, and Americas playing a role in the determination of transfer entropy?

5) Several of the references are appropriate and informative. However, other studies dealing with causality detection in financial markets are missing from the reference list. Authors might find a nice literature review in the paper Sandoval, L., 2014. Structure of a global network of financial companies based on transfer entropy. *Entropy*, 16(8), pp.4443-4482 and select from it the literature they consider appropriate.

Author's Response to Decision Letter for (RSOS-200863.R0)

See Appendix A.

Decision letter (RSOS-200863.R1)

Dear Professor Aste,

It is a pleasure to accept your manuscript entitled "Information-theoretic measures for non-linear causality detection: application to social media sentiment and cryptocurrency prices" in its current form for publication in Royal Society Open Science.

on behalf of Professor Manfred Broy (Associate Editor) and Marta Kwiatkowska (Subject Editor)
openscience@royalsociety.org

Appendix A

Manuscript Resubmission

Dear Editor,

please find enclosed the manuscript entitled “Information-theoretic measures for non-linear causality detection: application to social media sentiment and cryptocurrency prices” that we submit for publication as a regular article on Open Science.

We are thankful to the reviewers for the careful reading of the manuscript and their comments that helped to highly improve the quality of the manuscript.

We have revised the whole manuscript following their suggestions and criticisms. We accepted a large part of their suggestions and we modified the manuscript accordingly. Table 1 reports the reviewers’ comments and our modifications. We did not accept some of the minor suggestions and this is motivated in Table 2. A version of the manuscript with changes in blue is submitted as well to help verify the modifications to the manuscript.

We hope that the present version is judged acceptable for publication in your journal.

With kind regards,

Tomaso Aste and Keskin Zac

Changes in response to reviewer's comments

Please find the list of changes that have been made to the manuscript in grateful response to the very helpful feedback from reviewers. We split these up into required fixes and other comments. First points "that absolutely need fixing":

Table 1: Changes which must be made

Reviewer	Comment	Response / Action Taken	Location
Reviewer 1	You repeat in many places that sentiment causes /price/. This would be both statistically (non-stationarity) as well as economically (arbitrage) extremely worrying. Now, you do what should be done (take log differences, and say so in §2 of Sec 5), but the terms should be straightened out in the entire paper: you study the effects of changes in sentiment on returns! (and vice versa)	We acknowledge this was an area which required clarification - in particular at the top of p3 where we introduce the Granger causality test - and so we have clarified this throughout. In some cases it is reasonable from context that, if changes in sentiment cause changes in price, then price is in some way driven by sentiment.	Throughout manuscript
Reviewer 1	The "vice versa" is natural in the context of Granger causality and transfer entropy, but your liberal use of "causality" as synonymous with those will likely offend many theorists. It would certainly have offended Hume, your first reference. All this can still be fine if you declare this "equivalence" for the scope of your paper; however, when you (implicitly) dismiss other meanings of causality (including the standard and intuitive one), this goes too far: shuffling will destroy, in expectation, the time dependence you measure – and thus I have no objections to your method – but it will certainly not "hence [destroy] any causality"	We acknowledge that it was not specified that/where, when using the word causality, we are (usually) referring only to the statistical causality interpretation of Wiener / Granger. We have now replaced references to causality, to clarify where we mean statistical/G-causality	Throughout manuscript
Reviewer 1	You also over-reach in your statement on p 11 claiming to provide evidence of sub-hourly dependence patterns. You have indication of a stronger pattern as short as one-hour lags; there is no reason to conclude about frequencies you cannot study.	Yes acknowledged; the evidence is clear for hourly dependences and this is insufficient for concluding about higher-frequency relationships, whilst it is sufficient for demonstrating that there is information transfer at hourly timescales. We have removed these extraneous claims.	Conclusion section, p15.

We also received many helpful suggestions for areas which may provide additional context or colour to our findings. These are detailed in table 2 below, alongside our comments where applicable.

Table 2: Other points and minor comments

Reviewer	Comment	Response / Action Taken	Location
Reviewer 1	While you link well to the literature on sentiment analysis in crypto markets, you do not mention at all that your problem is at the heart of a large financial literature on price discovery	Indeed the search for alpha is contributory to price formation in financial markets, much in the way arbitrage is, however this is not key to the demonstration of meritorousness of information-theoretic techniques. Nevertheless the manuscript is improved by the mention of signals/alpha generation as a part of price discovery.	Introduction section, p2.
Reviewer 1	This would have also led you to the obvious question, do you find "causality" between returns of different coins?	We sought to avoid decomposing returns into multiple factors as the complexity of the problem can detract from the clarity/simplicity of the result when focusing on one driving time series. A virtue of both techniques used in the paper is that, though the causality detected may be due to one or more hidden variables, our results show that predictability of the price is increased by considering sentiment - which is sufficient.	
Reviewer 1	The paper would also profit from at least some discussion of why you believe the standard approach to handle non-linearities is insufficient, namely transforming the driver process. In relation to this, the fact that you find mostly no linear but non-linear dependence begs the question how the relation does look like. You should be able to estimate the relationship, ideally provide a graph.	Although this sounds interesting, as with the previous point we seek to address one source of information at a time, rather than decompose into (parametric) non-linear functional relationships. The true form of the dependence we detect would be interesting, albeit challenging, to reliably uncover, and we expect this is non-stationary in any case.	
Reviewer 1	The other elephant in the room which goes unmentioned is liquidity. You should have trading volume data from cryptocompare and could thus address this easily with at least the Amihud measure.	We consider liquidity effects, and other execution/transaction-related drivers, as factors which contribute to changes in price in concert with sentiment and other latent variables. We acknowledge that (il)liquidity may modulate or interrupt the form of the causality from sentiment to price, but in this paper we don't seek to decompose these factors. We are still able to ask the specific question \rightarrow "does knowing social sentiment data reduce the uncertainty in changes in price?", for which a positive result evidently holds despite the liquidity effects.	
Reviewer 1	Consequently, this raises the point of alternative drivers. Given that you have done your placebo tests on generated data, why not show what causality you can detect from LTC sentiment changes on BTC returns, etc?	As above, understanding the alternate drivers does not impact the results, but may allow us to apply a stricter condition. E.g. if LTC sentiment \rightarrow BTC price \rightarrow LTC price, then we can observe LTC sentiment \rightarrow LTC price, despite the effect of BTC (c.f. Pearl's work on causal graphs / Bayesian Networks)	

Reviewer	Comment	Response / Action Taken	Location
Reviewer 1	Finally a marginal point which may be subjective, but I do not buy the argument that we should expect more effect of sentiment in crypto markets due to their limited role as media of exchange. Common stock is much harder to exchange, and many more unsophisticated investors are investing in stock markets. I do not claim that the speculative motive is smaller; I just don't believe your chain of argument to be convincing in this regard. And you do not even need it in the least: sentiment analysis is a strong (if controversial) field in financial economics outside and inside of cryptocurrency markets, your question is valid and important independent of your beliefs about how cryptos "should" be priced.	We acknowledge this point; the justification is that where tangible value (e.g. dividends, inventory etc.) obviously contribute some % towards price formation in equities, for cryptocurrencies these do not apply so a larger % must come from other drivers e.g. sentiment. However the comment was speculative and has now been removed from the manuscript.	Introduction section, p2
Reviewer 2	The topics is timely and the authors develop it properly. The results are interesting for the interdisciplinary research community interested in quantitative modeling of financial market dynamics.	We appreciate the positive comments and agree on the interdisciplinary value of the results	
Reviewer 2	The authors consider prices of four cryptocurrencies. In several other studies of casualties asset returns have been considered. Why they choose prices? Would a study done with returns produce the same or similar results?	We think this confusion reflects our potentially confusing description of causality between sentiment and price, in contrast to the differencing procedure which means we technically calculate causality between changes in sentiment and returns. This was also pointed out by Reviewer 1, and we have now clarified this.	Throughout the manuscript
Reviewer 2	The authors are using 24-month windows with a one month step. Most of the results of figures 4-7 therefore are highly correlated at successive time records. To better understand the dynamics of the nonlinear causality it would be interesting to see how the results change when the overlapping between time windows decreases;	This is true and the results of each window are correlated; the effect of this is only seen in the tables presented in the Appendix, and we had already drawn attention to this fact in the caption. However we acknowledge it could be of interest to observe the results using windows with no overlap; the challenge is the number of windows becomes very small, or else the length becomes small and so the significance within each window is decreased. There is a trade-off and for this paper we think the approach taken in sensible.	

Reviewer	Comment	Response / Action Taken	Location
Reviewer 2	The recent literature on sentiment signals extracted from tweets highlights the evergrowing role of bots in the dynamics of Tweeter. Did the authors perform any pre-processing to evaluate the presence or absence of bot actions in the construction of their sentiment data series? Are bot present? Do they have any role?	The effect of individual agents is not considered, whether automated or human, and such microstructure/liquidity effects represents whole (fascinating but separate) topic within algorithmic trading. As commented elsewhere in this table, a key benefit of approach taken is that the decomposition of the causal relationships between sentiment and price is not required to show (and exploit) the observed relationships.	
Reviewer 2	Trading of cryptocurrency and Twitter activity are geographically constrained (although markets for cryptocurrencies are active 24 hours a day). Are intraday patterns due to geographical activity of Asia, Europe, and Americas playing a role in the determination of transfer entropy?	This is an interesting point where some linguistic / time-zone effect may come into play and contribute to information transfer over different timescales. However since both Twitter and Crypto markets are continuous 24/7, and since we look at relatively short intraday periods of 1 hour, it is reasonable to conclude these effects are minimal and do not contribute to the observed causal relationships.	
Reviewer 2	Several of the references are appropriate and informative. However, other studies dealing with causality detection in financial markets are missing from the reference list. Authors might find a nice literature review in the paper Sandoval, L., 2014. Structure of a global network of financial companies based on transfer entropy. Entropy, 16(8), pp.4443-4482 and select from it the literature they consider appropriate.	We appreciate the recommendation.